# Operationalizing Carbon-Neutral Living: A Case Study of A Business Model for Carbon-Negative Products

**Yue-Rong Hong [1], Chien-Ming Lee [2],\* and Tsai-Chi Kuo [3]**

1 Department of Information Management, National Taiwan University of Science and Technology, Taipei 106, Taiwan; greenart1010@gmail.com
2 Institute of Natural Resources Management, National Taipei University, Taipei 237, Taiwan
3 Department of Industrial Management, National Taiwan University of Science and Technology, Taipei 106, Taiwan; tckuo@mail.ntust.edu.tw
\* Correspondence: cmlee@mail.ntpu.edu.tw; Tel.: +886-2-8674-1111 (ext. 67335); Fax: +886-2-8671-6313

**Abstract:** Behavior change is the last hurdle to achieving net zero emissions by 2050. However, if the public does not take responsibility for emissions that arise from their life activities, low-carbon behavioral changes are unlikely to be adopted. This article introduces personal social responsibility (PSR), and advocates that individuals should make good use of consumer sovereignty and wise consumption power. This study aims to investigate the feasibility of the carbon-negative business model and, accordingly, the opportunity to practice carbon neutrality in personal life. This article uses carbon-negative commodities as a carrier to establish a new model of personal carbon trading (PCT), and introduces behavioral nudge design to encourage the public to adopt low-carbon behavior. Regression analysis shows that the sales volume of carbon-negative products and the cumulative number of members are significantly related (10% and 1% significance). This indicates that attaching 1 kg of carbon credits (costing about TWD 0.5) can stimulate sales of a carbon-negative product to increase by 1.6, making it economically feasible. This verifies that the public can accumulate carbon assets through the consumption of carbon-negative products, offset carbon emissions from daily life, and gain the opportunity to practice carbon-neutral life. Product carbon footprint labeling is the basis of this pilot project; therefore, promoting product carbon footprint labels is needed, and is recommended to ensure a successful carbon-neutral living transition.

**Keywords:** net zero emissions; carbon-negative products; personal carbon trading; personal social responsibility; carbon footprint

## 1. Introduction

It is evident that carbon neutrality has become the most important issue and challenge of this century for humankind to respond to climate change. However, the global target of greenhouse gas emissions pledges would need to be six times higher to keep the global temperature rise below 2 °C above pre-industrial levels and seven times higher to limit warming to 1.5 °C [1]. This indicates the failure to respond to extreme weather and climate change as the greatest risk to humanity [2]. Insufficient global participation and carbon price signals are currently the most pressing challenges in the global response to climate change [3]. Thus, expanding participation and strengthening carbon price signals are crucial tasks to address climate change. Consumer demand is the main determinant of production by enterprises (the theory of consumer sovereignty). Then, the key issue is how to awaken the public and stimulate action. Promoting low-carbon consumption is thus one of the most effective strategies to control carbon emissions from production by enterprises. Consequently, the United Nations Framework Convention on Climate Change (UNFCCC) has promoted the Momentum for Change initiative to encourage people to make small changes that can create a global momentum for reductions in greenhouse gas emissions.

Behavior change (low-carbon life), on average, could contribute about 10–15% of global greenhouse gas emission reductions annually [4]. Vendenbergh and Steinemann [5] draw on norms theory and empirical studies to demonstrate how legal reforms can tie the widely held abstract norm of personal responsibility to the emerging concrete norm of carbon neutrality. If people are to change their carbon-emitting habits and behavior, it is critical to identify solutions enabling capability, motivation, and opportunity [6]; this means that behavior occurs as an interaction between three conditions [7]. It can be seen that grasping the behavior characteristics of the people's bounded rationality, by designing a nudge mechanism, will be an important issue to promote behavior change.

Behavioral science/economics was developed in the 1950s to explain the decision-making inertia of humans or economic individuals, mainly based on the following theoretical foundations: (1) the "loss aversion" preference that emphasizes loss [8]; (2) the "certainty effect", to select a certain result [9]; (3) the "framing effect" [10], where the choice depends on the presentation; (4) the "present bias" [11]: pay attention to the present small profits (or neglect the big profits in the future); (5) the "peer effect" (or social preference) [12], describing susceptibility to peer influence (or pressure); (6) the "endowment effect" [13], which overestimates the value of one's own assets; (7) the heuristics [14], where people make decisions in a simple and intuitive way (making decisions based on past experience is prone to system bias). These findings can show how to apply the concept of 'nudging' and design the choice environment to prompt individuals to make choices that are aligned with their stated intentions for practicing carbon-neutral living.

Personal carbon trading (PCT) is one of the most effective ways for generating "good momentum for change" in society, promoting low-carbon consumption behavior and achieving global goals to reduce greenhouse gas emissions through the allocation of personal carbon emissions [15–17]. In recent years, there has been an abundance of literature on the application of the PCT to promote the transformation of low-carbon transportation for the public [18–24]. There is a wealth of literature on institutional design [25–27]. However, its lack of acceptance in society [15,17,24,28,29] and the high costs of policy implementation are the main constraints to its adoption [30–32]. PCT incorporates personal carbon emission management, demand-side drivers, and the change of social norms; the main benefits of PCT are argued to be innovative and comprehensive carbon reduction methods [27]. Moreover, PCT is a more effective policy tool than carbon taxation [18,27,33,34]. However, the traditional literature discussing PCT schemes is based on caps on allowance allocation [35]. This means that PCT will face extremely high enforcement costs, limiting its development. In short, reducing transaction costs and acceptance by the public is a key issue for achieving the success of PCT [21].

PCT refers to the acquisition of carbon credits by individuals through the trading of the carbon assets attached to carbon-neutral or carbon-negative products. Carbon-negative products are products for which their carbon footprint is lower than their carbon credits. In other words, a surplus remains after the carbon dioxide emissions during the product life cycle are offset by the carbon credits generated through carbon abatement activities. For example, if the carbon footprint of a product is 0.728 kg, after offsetting the carbon footprint with 1 kg of carbon credits, there is a surplus of 0.272 kg. When compared with policies such as taxes, fees, and direct controls, PCT mechanisms have the advantages of public engagement, environmental effectiveness, and fairness [31]. In addition, with appropriate institutional design and support measures, implementation costs can be reduced and public acceptability increased, which is conducive to green growth and the development of a low-carbon society. According to the OECD (2011) [36] definition, "Green growth means fostering economic growth and development, while ensuring that natural assets continue to provide the resources and environmental services on which our well-being relies". In this definition, green growth emphasizes not only "green" but also "growth". The former includes low-carbon, resource-efficient, and environmentally friendly production and consumption; the latter includes thinking about how to create momentum for sustainable growth, which means creating opportunities for future development. Following the Paris

Agreement, PCT is expected to become the most cost-effective and environmentally effective institutional mechanism for a low-carbon global society. Cost-effectiveness refers to the ability to achieve specific environmental goals at the lowest possible cost. Satoh [36] used radio frequency identification (RFID) tags to attach carbon credits to specific products. On purchasing a product, consumers also obtain the carbon credits attached to the product. This encourages purchase intention, creating a low-carbon production and consumption business model. However, Satoh [37] overlooked the carbon footprint of the product's production and therefore could not guarantee that the product would achieve net zero emissions (carbon neutrality). Carbon footprint refers to the greenhouse gas emissions during a product's life cycle.

Behavior change is the last hurdle to achieving net zero emissions by 2050. However, if the public does not take responsibility for emissions that arise from their life activities, low-carbon behavioral changes are unlikely to be adopted. Applied nudges to change people's energy-saving and low-carbon behaviors have been proven [38–40]. ERP [6] concluded some key issues for how behavior change will unlock net zero: (1) A sophisticated mixed of regulation, incentives, nudges, and penalties will be needed to motivate customers and industry towards net zero; (2) Policy-driven behavior change must be preceded with enabling plans for business and the general public to move towards a low-carbon lifestyle; (3) Carbon footprint information must be easily available to all age groups; (4) Further programs relating to net zero must consider and address barriers to behavior change across commercial and residential setting. Based on the above, this study cooperates with Taiwan's second largest convenience store group (FamilyMart convenience store) and to develop a new mindset and paradigm to encourage personal social responsibility (PSR), applying the concept of 'nudging', and designing the choice environment to prompt individuals to make choices that are aligned with their stated intentions for practicing carbon-neutral living. Research has found that attaching carbon credits to products does create incentives for consumers to purchase carbon-negative products.

This study aims to analyses the economic impact of carbon-negative commodities in PCT on various nudge design on convenience stores in Taiwan and a pilot store in the New Taipei city. The article mainly refers to the research of Satoh [36], and gives appropriate modification and expansion. The research establishes the optimal decision theory model of PCT for obtaining the amount of carbon credit attachment to the carbon-negative products. Then, based on behavioral science, this study designed a series of behavioral nudge environments, such as personal carbon accounts, carbon-negative products, and related incentive activities, etc. Further, this study collects sales data over three years (2020–2022) to investigate the feasibility of the carbon-negative business model, and accordingly, the opportunity to practice carbon neutrality in personal life.

The study is organized as follows: Section 1 is the introduction. Section 2 builds up a theoretical model of personal carbon trading. Section 3 constructs a business model for carbon-negative products. The results are discussed in Section 4 and conclusions presented in Section 5.

## 2. Theoretical Model of the PCT

This article assumes that a representative individual, on the basis of pursuing carbon-neutral living, accumulates personal carbon assets through the purchase of carbon-negative products. As representative firms pursuing the maximum profit are considered, it is necessary to decide how much carbon credit to attach to the products so as to promote the incentives for representative individuals to purchase carbon-negative products.

### 2.1. Optimal Solution

The representative consumer is faced with two types of product in the market: product with carbon credits and product without carbon credits. The representative consumer seeks to maximize utility through the optimal decision as follows:

$$\underset{(X,Y)}{Max} U(X,Y)$$

$$s.t.\ I_0 = P_X(1-\theta)X + P_Y Y$$

$$\theta = P^T(\bar{a} - \alpha X) \tag{1}$$

where $U$ is the utility function of the representative consumer, influenced by the consumption of the two products ($X$ and $Y$); the utility function is assumed to be a quasi-concave function, i.e., $U_X = \partial U/\partial X > 0$, $U_Y = \partial U/\partial Y > 0$, $U_{XX} = \partial^2 U/\partial X^2 \leq 0$, $U_{YY} = \partial^2 U/\partial Y^2 \leq 0$; $P_X$ and $P_Y$ are the product values, which are assumed to be fixed; $\bar{a}$ is the offset represented by a carbon credit, which is also assumed to be fixed (e.g., 1 kg); $P^T$ is the price of carbon credits, assumed to be fixed; $\alpha$ and $\beta$ are the carbon dioxide emission factors for product $X$ and product $Y$, respectively, also assumed to be fixed. Therefore, the carbon footprints of product $X$ and product $Y$ are $\alpha X$ and $\beta Y$, respectively. Assuming $\bar{a} > \alpha X$, $\theta = P^T(\bar{a} - \alpha X)$ therefore is the value of the carbon assets obtained by the consumer from purchasing product $X$ with attached carbon credits.

The optimal solution to the above problem is:

$$\frac{U_X}{\lambda} = P_X[1 - P^T(\bar{a} - 2\alpha X)] \tag{2}$$

$$\frac{U_Y}{\lambda} = P_Y \tag{3}$$

where $\lambda$ is the Lagrangian multiplier or the marginal utility of income. To simplify the analysis, assuming that the marginal utility of money is equal to 1, $\lambda = 1$, Equations (2) and (3) are reduced to:

$$U_X = P_X\left[1 - P^T(\bar{a} - 2\alpha X)\right] \tag{4}$$

$$U_Y = P_Y \tag{5}$$

From Equations (4) and (5), the product demand function can be solved:

$$X^D = X^D(P_X, P_Y, I_0, P^T, \alpha, \bar{a});\ Y^D = Y^D(P_X, P_Y, I_0, P^T, \alpha, \bar{a}).$$

### 2.2. Discussion

From Equations (4) and (5), we find that the consumer equilibrium is where the marginal utility ratio of the two products is equal to their relative prices, which is $U_X/U_Y = P_X(1 - P^T(\bar{a} - 2\alpha X))/P_Y$. From this, we find that compared to a situation where there is no carbon credit label, the real price of product $X$ is expressed as $\bar{a} - 2\alpha X$; in other words, whether the attached carbon credits more than double the carbon footprint of product $X$. This is the main influence factor. Three scenarios are discussed below.

Scenario 1: $\bar{a} - 2\alpha X = 0$

If the attached carbon credits are exactly twice the carbon footprint of product X, then the right-hand side of the Equation (4) is reduced to $P_X$, implying that the relative price of product X in terms of product Y does not change ($P_X/P_Y$). Therefore, the optimal decision of the representative consumer will not change.

Scenario 2: $\bar{a} - 2\alpha X > 0$

If the attached carbon credits are more than twice the carbon footprint of product X, this implies that the relative price of product X in terms of product Y is reduced. That is, as shown in Figure 1, the budget line rotates counterclockwise on the vertical axis, $P_X(1 - P^T(\bar{a} - 2\alpha X))/P_Y < P_X/P_Y$ reaching a new equilibrium point, represented by point

b, indicating that consumers will increase their purchase of product X, while changes in their purchase of product Y are uncertain.

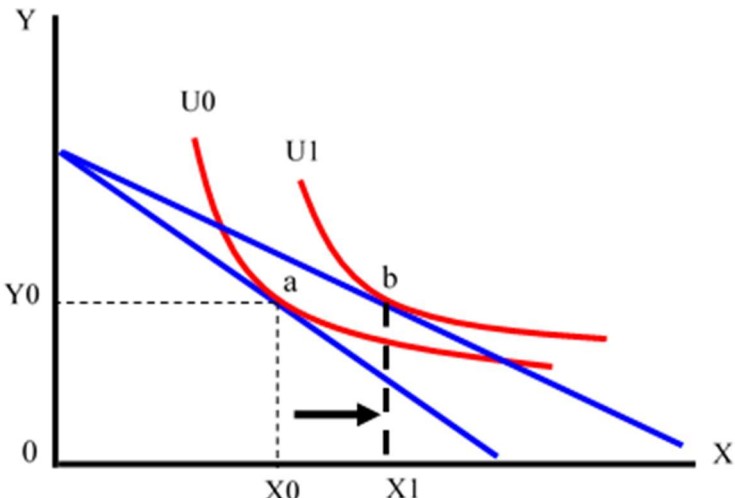

**Figure 1.** Attached carbon credits are more than twice the carbon footprint of product X, increasing consumption of product X.

Scenario 3: $\bar{a} - 2\alpha X < 0$

If the attached carbon credits are less than twice the carbon footprint of product X, it implies that the relative price of product X in terms of product Y is increased. That is, as shown in Figure 2, the budget line rotates clockwise on the vertical axis, $P_X(1 - P^T(\bar{a} - 2\alpha X))/P_Y > P_X/P_Y$, reaching a new equilibrium point represented by point b, indicating that consumers will decrease their purchase of product X, while changes in their purchase of product Y are uncertain.

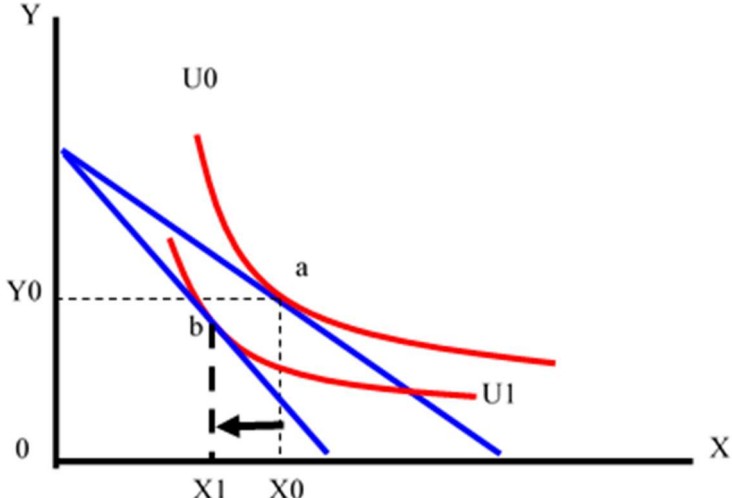

**Figure 2.** Attached carbon credits are less than twice the carbon footprint of product X, decreasing consumption of product X.

From the above discussion on economic implications, we find that attaching carbon credits is a necessary condition to induce consumers to increase purchases. When the number of attached carbon credits ($\bar{a}$) is more than twice the carbon footprint of product X, this is a sufficient condition to induce consumers to purchase products with carbon credit labeling.

### 3. Experimental Design

Lee et al. [41] conducted a survey of the preferences and willingness to pay for carbon-negative products of 411 residents from Sanxia District in New Taipei City. They found that 54% of respondents are willing to pay a higher price to purchase carbon-negative products, 53% are willing to pay a 5% higher price to purchase carbon-negative products, and 58% are willing to obtain carbon credit in exchange for their purchases of carbon-negative products. These results demonstrate the feasibility of low-carbon activities.

The research was conducted in collaboration with six FamilyMart convenience stores located in Sanxia District, New Taipei City. The carbon-negative products zone in the FamilyMart convenience stores is shown see in Figure 3. This study developed carbon-neutral product labeling to convey the message of carbon-negative consumption behavior; advocated lifetime personal carbon-neutral consumption for establishing new low-carbon living paradigm, i.e., promoted personal social responsibility (PSR), to form new social norms; as well as established a personal carbon asset account to create a friendly and low transaction cost environment to nudge consumer low-carbon behavior change.

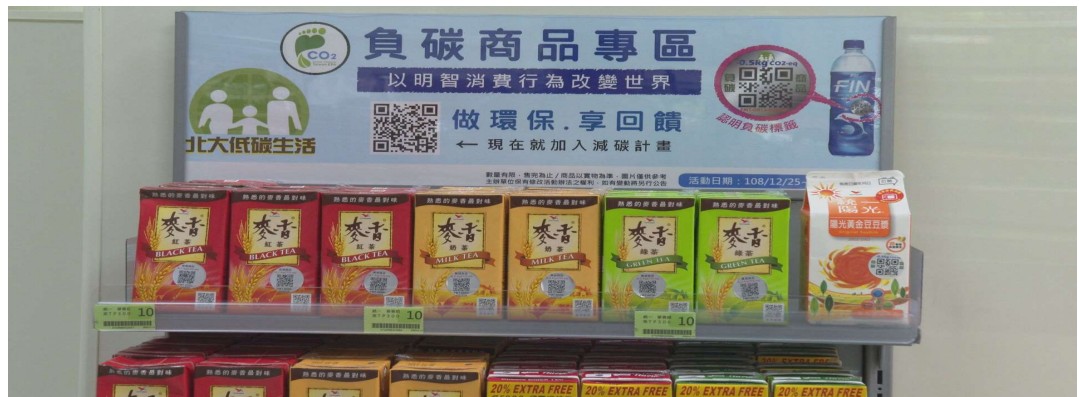

**Figure 3.** The carbon-negative product zone. Note: The non-English term means "The carbon-negative product zone".

#### 3.1. Nudging Experimental Design

Nudging design includes: (1) Deliver information by mobile phone (heuristic and reminder). (2) Establish a personal carbon account (default value). (3) Attached carbon credits are twice the product carbon footprint (default value). (4) In addition, 10 kg of carbon credits accumulate into TWD 10 discount coupons (default value). (5) "Carbon-negative commodity zone" (framework effect). (6) Timely carbon credits delivery and accounting information (present bias). (7) Low-carbon living outstanding reward (peer effect or social norm).

#### 3.2. The IoTs System of Carbon Assets Account

This study selects 20 products with carbon footprint labels, and then attaches twice the carbon credits This pilot purchased 100 tCO2e Gold Standard carbon credits that were obtained from Taiwan's wind power generation. The Gold Standard was created by certain organizations, including the World Wide Fund for Nature, SouthSouthNorth Initiative, and HELIO International, and formally launched in 2003 after a long process of consultation with stakeholders, including government departments, environmental agencies, private companies (including investors and project developers), and certification bodies. According to previous theoretical results, this forms carbon-negative commodities (QR-code tag, see Figure 4). Then, if an individual purchases a carbon-negative product labeled with a 300 g carbon footprint with 600 g of carbon credits attached, then 300 g carbon credit offset the carbon footprint; the product becomes a carbon neutral product, and the remaining 300 g

will transfer to the individual's carbon asset account to encourage his/her responsible consumption behavior, thus completing personal carbon trading (PCT).

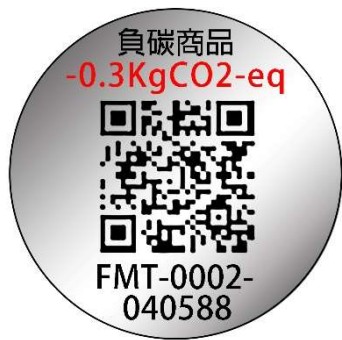

**Figure 4.** The carbon-negative product QR code. Note: The non-English term means "Carbon-negative product".

Internet of Things (IoTs) technology is applied to uniquely identify and trace the carbon credits acquired through an individual's product transactions (or PCT), and also provide carbon credit account management services (personal carbon asset account, PCAA) for trading, transferring, and storing individual carbon credits. The operation procedure is described as follows (see Figure 5): (1) Step 1: establishing a personal carbon asset account. (2) Step 2: completing a carbon-negative product transaction. (3) Step 3: consumers scan the QR-code on the tag of carbon-negative products; then, the data will be sent to the Low Carbon Life Net server. (4) Step 4: The FamilyMart server will send the transaction data of the consumption to the server of Low Carbon Life Net at the same time and check the data to confirm that it is correct, then the carbon credit delivery will be completed. The next day, the system will notify consumers by email that the carbon credits have been entered into the PCAA.

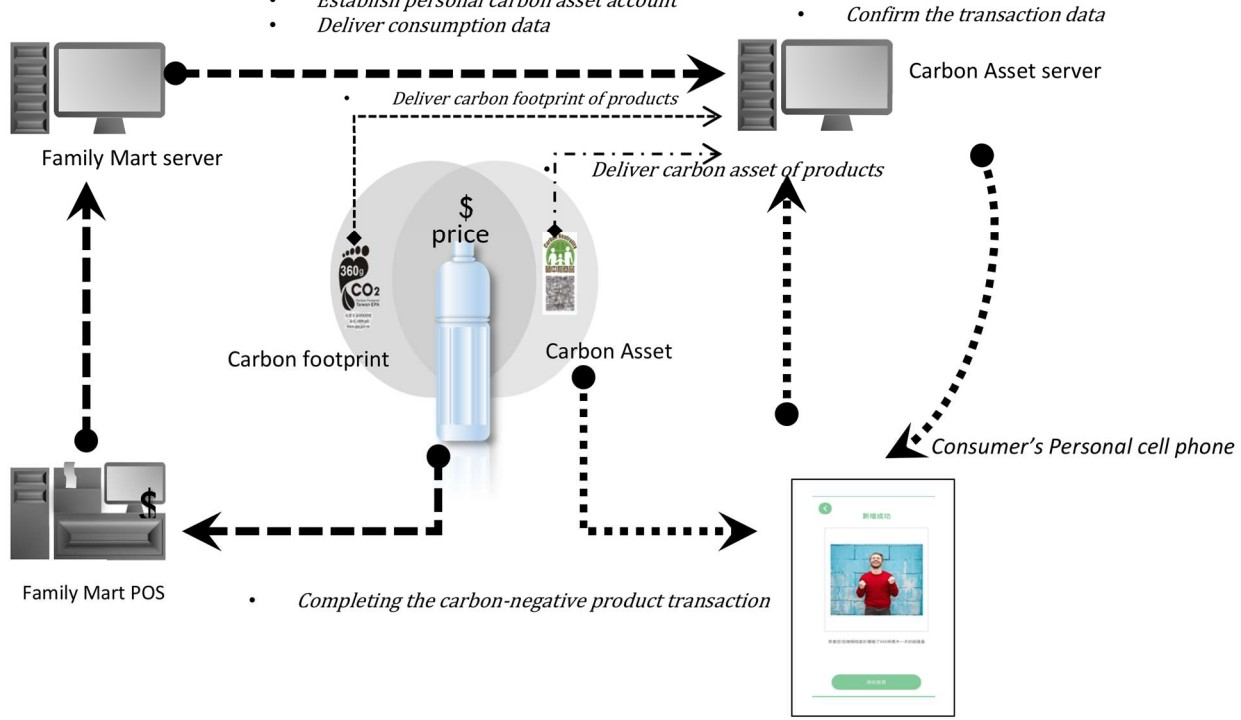

**Figure 5.** The carbon assets information system.

### 3.3. Personal Social Responsibility Initiative

Per capita greenhouse gas emissions have been increasing year by year, reaching approximately 10.92 tons CO2e in 2020 in Taiwan [42]. If current consumption patterns continue, it will be difficult to achieve the goal of carbon neutrality by 2050. Therefore, creating a new value of becoming lifelong carbon neutral has become an important issue for low-carbon life transition; that is, encouraging the public to reduce their carbon emissions and to offset such emissions through the purchase of carbon-negative products, in order to achieve ultimately the goal of lifetime carbon neutrality.

From the perspective of consumption-based greenhouse gas emissions calculations, consumers should take responsibility for reducing greenhouse gas emissions. Therefore, how to make good use of consumer sovereignty to influence low-carbon production activities will become a key issue. Then, a new social paradigm of "personal social responsibility" (PSR) (see Figure 6) in the form of a "personal carbon budget" should be established, encouraging individuals to consume wisely (carbon-free or negative-carbon consumption) to drive the transition to the production of carbon-free or carbon-negative activities, ultimately achieving the aim of carbon-neutral living.

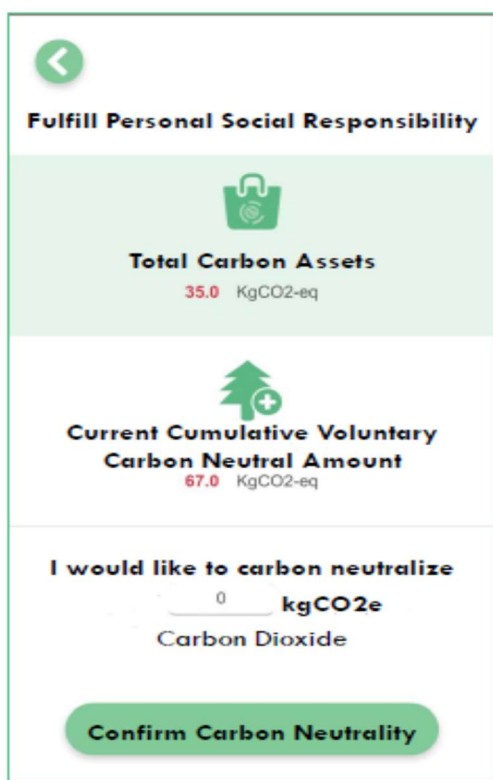

**Figure 6.** Personal carbon credit account.

## 4. Results and Discussion

This pilot program was launched in December 2019. During the implementing period (2020–2022), there was only one carbon-negative store in 2020, a second carbon-negative store was added in 2021, and three more carbon-negative stores were added in 2022, for a total of five stores. This period coincided with COVID-19, which affected the normal operation of convenience stores.

In three years, this scheme has accumulated over 1900 members and sold more than 100,000 carbon-negative products. This study first explains the cumulative sales, membership accumulation, and carbon credit attachment of 20 carbon-negative products over the past three years. Further, the first quarter of 2019 was used as the baseline to compare the sales of carbon-negative products in the first quarter of three consecutive years (i.e., 2019Q1, 2020Q1,

2021Q1, 2022Q1), and to provide considerable empirical evidence to support whether the experimental nudge design really changed people's low-carbon consumption behavior.

### 4.1. Sales Volume of Carbon-Negative Products

The study collected the sales volume of 20 carbon-negative products in 2019Q1 (1324), 2020Q1 (1325), 2021Q1 (1387), and 2022Q1 (2460). See Figure 7 for details. This study takes 2019 as the baseline sales volume, and it can be seen from Figure 7 that the sales volume of carbon-negative products has shown continuous growth for three consecutive years.

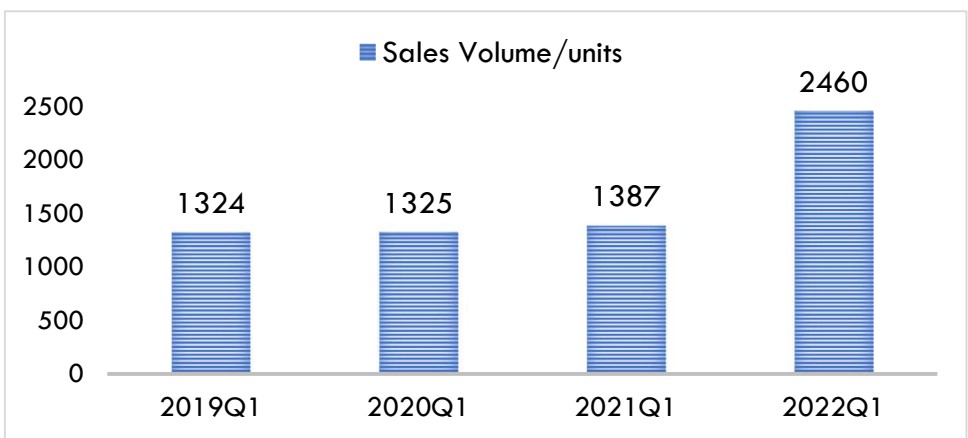

**Figure 7.** Sales volume (units) from 2019Q1 to 2022Q1.

The sales volume in 2022Q1, which increased significantly, is estimated to be affected by three factors. One is that the public has been gently nudged by carbon-negative products; another is that the COVID-19 epidemic has gradually unlocked; the third is the increase in participants.

### 4.2. Regression Results

This research attempts to demonstrate whether products with carbon-negative labels (or with carbon assets) could significantly nudge people to buy carbon-negative products. The regression equation is as follows:

$$Y = a + bX_1 + cX_2 + e$$

where $Y$ is monthly sales volume of 20 carbon-negative products; $X_1$ is the monthly carbon credits attached to 20 carbon-negative products; and $X_2$ is the amount of participants accumulated. The data are shown in Table 1.

**Table 1.** Data collection of regression.

| Sales Volume (Units) /Month | Carbon Credits Attachment (g)/Month | Cumulative Membership (Person)/Month |
|---|---|---|
| 448 | 136,600 | 149 |
| 275 | 27,200 | 225 |
| 609 | 61,000 | 285 |
| 604 | 24,800 | 288 |
| 969 | 81,000 | 326 |
| 1177 | 122,800 | 346 |
| 77 | 8800 | 350 |
| 803 | 12,800 | 357 |

**Table 1.** *Cont.*

| Sales Volume (Units) /Month | Carbon Credits Attachment (g)/Month | Cumulative Membership (Person)/Month |
|---|---|---|
| 1049 | 31,400 | 359 |
| 1256 | 53,800 | 450 |
| 937 | 21,400 | 477 |
| 270 | 10,400 | 484 |
| 209 | 11,800 | 484 |
| 908 | 29,000 | 491 |
| 980 | 60,200 | 498 |
| 732 | 22,800 | 501 |
| 159 | 10,400 | 503 |
| 61 | 2600 | 508 |
| 499 | 15,000 | 516 |
| 1137 | 62,600 | 869 |
| 1352 | 45,000 | 1040 |
| 660 | 16,400 | 1060 |
| 288 | 4800 | 1061 |
| 1512 | 23,600 | 1090 |
| 859 | 1800 | 1091 |
| 332 | 4200 | 1092 |
| 4053 | 221,400 | 1410 |
| 6104 | 3200 | 1427 |
| 6174 | 39,600 | 1598 |
| 6351 | 41,600 | 1693 |
| 5340 | 24,400 | 1741 |
| 3549 | 4600 | 1751 |

For details of the regression results, see Table 2. It shows that the amount of carbon credits attached has a 10% significance to the sales of carbon-negative products; the accumulation amount of participants has a 1% significance to the sales of carbon-negative products. The economic implication of the $X_1$ regression coefficient 1.6 points out that compared with the baseline sales volume (the same product in 2019Q1, but without carbon credit attached, that is, the sales volume of non-carbon-negative products), the sales volume of products with carbon credits (i.e., carbon-negative products) 60% increase. This result verifies that carbon-negative products do have the effect of nudging the public to change their purchasing behavior, and corroborates the questionnaire survey results of Lee et al. [41]. More importantly, during the implementation period, although it coincided with the unfavorable environment of COVID-19, the sales of carbon-negative products were robust, showing climate resilience.

The regression coefficient of the cumulative number of participants is 3.12, which means that every additional participant will add 3.12 carbon-negative products, implying that expanding public participation will be the core issue of promoting carbon negative business models. This pilot program has accumulated more than 1900 participants, and will continue to encourage local people and communities to participate and build a more convenient and friendly environment for participation.

**Table 2.** Regression results of carbon credit attachment and accumulative memberships.

| Parameter | Estimated Value | t-Value | *p*-Value | R2 |
|---|---|---|---|---|
| Slope | −1142.76 *** | 2.79 | <0.01 | |
| Carbon Credit Attachment | 1.60 * | 1.86 | 0.07 | 0.681 |
| Participants Accumulation | 3.12 *** | 7.73 | <0.01 | |

Note: *** indicates 1% significant level; * indicates 10% significant level.

## 5. Conclusions

The remaining carbon budget for limiting global warming to 1.5 °C will likely be exhausted within this decade; however, economic policy instruments to combat this are potentially very costly [43]. The era of climate emergency is coming and climate action failure is a major risk for the next decade. We should recognize that the current production-based climate policy is not the real solution to the climate crisis. Therefore, a breakthrough in the climate dilemma will be at stake if the public are included in climate policy. In other words, we need to change to consumption-based greenhouse gas emission responsibilities, and implement the responsible consumption of SDG12, influence responsible production, and form effective climate action then fulfil SDG13.

Everyone should have a carbon asset account to endow the public with low-carbon behavior value, and recognize that carbon assets are genuine wealth, which will act as an important nudge to promote the low-carbon transition of the public. This study starts from consumption-based GHG emission responsibilities; we then propose a carbon-negative products business model to provide the basis for widely applied nudging design and PCT systems to facilitate carbon-neutral living. This study collects three years of data for the consumption of twenty carbon-negative commodities, and applied the behavioral science EAST (East-Attractive-Social-Timely) nudging design, such as mobile phone APP (easy), carbon-negative product zone (attractive), personal carbon asset account (PCAA) (attractive), cash discount (10 kg offset 10 cash) (attractive), and email return of results (timely).

This study attempts to quantify the research results. The empirical results show that carbon-negative products will indeed attract people to buy them. Compared with baseline products, the sales potential will increase by 60% on average. The sales potential of carbon-negative products shows that attracting public participation will be a key factor in promoting carbon-negative business models. This research verifies the feasibility of the carbon-negative business model. We conclude that PCT and personal carbon budget will need to become an integral part of the global climate policy mix if we are to ensure the viability of ambitious climate targets and the equitable distribution of mitigation efforts across individuals.

Product carbon footprint labeling is the basis of this pilot project. According to this, the amount of attached carbon credits and the amount of carbon assets allocated to consumers can be correctly calculated. Therefore, the promotion of carbon footprint labels is not only empowering, but also boosts important policy measures for carbon-neutral living.

Behavior change is a long-term phenomenon. This study only observed 3 years of data. The preliminary research results still need continuous observation. This is the first limitation of this study. The experimental sample of this study is about 1900 people and 20 carbon-negative commodity items. More experimental samples need to be expanded to obtain more robust results, which is another limitation of this study. This study verifies that appropriate nudge design can indeed change people's behavior. In future, it can be further extended to transportation and other energy-saving behavior change experiments.

**Author Contributions:** Conceptualization, Y.-R.H. and C.-M.L.; Methodology, C.-M.L.; Software, Y.-R.H. and T.-C.K.; Validation, Y.-R.H. and C.-M.L.; Formal analysis, Y.-R.H. and C.-M.L.; Investigation, C.-M.L. and T.-C.K.; Writing—original draft, Y.-R.H.; Writing—review and editing, C.-M.L. and T.-C.K.; Visualization, Y.-R.H.; Supervision, C.-M.L. and T.-C.K. All authors have read and agreed to the published version of the manuscript.

**Funding:** This research was partially funded by National Taipei University and Ministry of Education.

**Institutional Review Board Statement:** The study was conducted according to the guidelines.

**Informed Consent Statement:** Informed consent was obtained from all subjects involved in the study.

**Data Availability Statement:** The data presented in this study are openly available in reference number.

**Conflicts of Interest:** The authors declare no conflict of interest.

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
