# Peer review of "Operationalizing Carbon-Neutral Living: A Case Study of A Business Model for Carbon-Negative Products"

_sustainability, doi:10.3390/su151411315_

Round 1
Reviewer 1 Report
The topic of this article is interesting but there are many issues regarding the writing and the discussion on the results. The detail of my notes is as follows:
1) Please check the grammar thoroughly and fix the errors. For example, there are issues concerning the writing in the abstract:
.... and to develop a new mindset and paradigm to encourage personal social responsibility (PSR), applying the concept of ‘nudging’ and design the choice environment ....
.... the carbon-negative products has a significant grew, this indicates that the feasibility of the carbon negative business model, and the opportunity for net zero living transition.
2) The current citation format is uncommon. Writing only the citation number is enough; do not mix them with the author(s) name and year of publication.
3) Typos and other concerns regarding the writing:
· Typo in line 56: the words “certainty effect” is written twice.
· In line 248, it is mentioned that “The pilot program was launched in December 2019” but in line 258 it is mentioned that “The pilot project was launched in 2020Q1”. These statements are confusing.
· Line 260-261: The authors mention that Figure 1 shows the sales growth. Is it correct “Figure 1”?
· Typo in line 265: cocid-19
· Typo in line 273: “sales volume of” is written twice
· Typo in line 279. The title of Table 1 is unclear
· The first column of Table 1 is unclear. Is it sales volume per month?
4) The research questions, research objectives, and main contributions of the paper should be stated clearly at the end of Section 1.
5) The discussion on the results of the analysis, especially in Section 4, is weak. The implications and practical/business insights should be discussed. Does the result verify the feasibility of the business model? Connect the results with existing theories and the results of previous studies.
6) Which part of the results supports your conclusion “PCT and personal carbon budget will need to become an integral part of the global climate policy mix”?
7) The limitations of this research and future extension should be explained in the conclusion.
There are many issues regarding the writing including some typos and grammar, check the complete comments.
Author Response
Sustainability - 2427739
Response to Reviewer 1 Comments
Point 1: Please check the grammar thoroughly and fix the errors. For example, there are issues concerning the writing in the abstract:
(1)... and to develop a new mindset and paradigm to encourage personal social responsibility (PSR), applying the concept of ‘nudging’ and design the choice environment ....
(2) …the carbon-negative products has a significant grew, this indicates that the feasibility of the carbon negative business model, and the opportunity for net zero living transition…
Response 1: Thank you for your comment. The authors have rewritten abstract. (Please see abstract)
Point 2: The current citation format is uncommon. Writing only the citation number is enough; do not mix them with the author(s) name and year of publication.
Response 2: Thank you for your comment. The authors have reformatted according to journal format. (Please see article)
Point 3: Typos and other concerns regarding the writing:
- Typo in line 56: the words “certainty effect” is written twice.
- In line 248, it is mentioned that “The pilot program was launched in December 2019” but in line 258 it is mentioned that “The pilot project was launched in 2020Q1”. These statements are confusing.
- Line 260-261: The authors mention that Figure 1 shows the sales growth. Is it correct “Figure 1”?
- Typo in line 265: cocid-19
- Typo in line 273: “sales volume of” is written twice
- Typo in line 279. The title of Table 1 is unclear
- The first column of Table 1 is unclear. Is it sales volume per month?
Response 3: Thank you for your comment. Typos modification as follows:
- Remove duplicates. (please see line 60 red words)
- Delete the sentence “This pilot project was launched in 2020Q1”.
- Corrected to Figure 7. (please see line 289 red words)
- Corrected to covid-19. (please see line 294 red words)
- Remove duplicates. (please see line 304 red words)
- Corrected to” data collection of regression” (please see line 325 red words)
- Corrected to sales volume per month. (please see line 325 red words)
Point 4: The research questions, research objectives, and main contributions of the paper should be stated clearly at the end of Section 1.
Response 4: Thank you for your comment. The authors have added a paragraph to describe the research method and the materials used of the study in introduction. (Please see line 113-122 red words).
Point 5: The discussion on the results of the analysis, especially in Section 4, is weak. The implications and practical/business insights should be discussed. Does the result verify the feasibility of the business model? Connect the results with existing theories and the results of previous studies.
Response 5: Thank you for your comment. The authors have enhanced the interpretation of the results. (Please see line273-285, and line 307-324 red words.)
Point 6: Which part of the results supports your conclusion “PCT and personal carbon budget will need to become an integral part of the global climate policy mix”?
Response 6: Thank you for your comment. The authors have expanded conclusions section for pointing out the outlines of this article. (Please see line 333-372)
Point 7: The limitations of this research and future extension should be explained in the conclusion.
Response 7: Thank you for your comment. The authors have added further expansion and research limitation of this study to the last paragraph of the concluding section. (Please see line 366-371 red words)

Reviewer 2 Report
The topic that the authors are exploring is very interesting and also very important for the long-term future of humanity. Environmental degradation on an unprecedented scale is placing a huge burden on the environment, the serious consequences of which are already well visible.
However, despite the interesting subject matter, the article needs to be improved on a number of points. The following introduction, which is also a literature review, does not, in my opinion, provide a sufficient basis for the topic under consideration and I would therefore recommend that it be expanded. I would also recommend that the number of references used be increased. An article is good writing if either the literature or the research is strong. In this article, I feel that neither is strong. Having said that, for me the structure of the article is very mixed. I do not see the research method and the materials used properly outlined. The mathematical foundation seems to be there, but it would be good to put this in a Materials and Methods chapter. I have the same opinion about the results section. There are very few explanations that outline what the study has to say, but there are many figures and pictures and tables, which are left to the reader to interpret. I would also recommend that the conclusions section be expanded, as it is insufficient. So after considerable revision, I recommend the article for publication.
Author Response
The authors divide the above questions into the following 3 points:
Point 1: Which is also a literature review, does not, in my opinion, provide a sufficient basis for the topic under consideration and I would therefore recommend that it be expanded.
Response 1: Thank you for your recommendation. PCT of this study is different from the traditional literatures, mainly referring to the research of Satoh (2014), and giving appropriate modification and expansion. The authors have expanded more related literature review in the Introduction. (Please see line 77-84 red words)
Point 2: The structure of the article is very mixed. I do not see the research method and the materials used properly outlined.
Response 2: Thank you for your comment. The authors have added a paragraph to describe the research method and the materials used of the study in introduction. (Please see line 113-122 and line 128-132 red words)
Point 3: I would also recommend that the conclusions section be expanded, as it is insufficient.
Response 3: Thank you for your comment. The authors have expanded conclusions section for pointing out the outlines of this article. (Please see line 333-372 red words)

Round 2
Reviewer 1 Report
Thanks, the authors have addressed my previous comments.
Author Response
Point 1: Thanks, the authors have addressed my previous comments.
Response 1: Thank you for your valuable comment.
Reviewer 2 Report
The study has improved a lot since its inception and now looks much more scientific and professional. For me, the underlying literature is still incomplete and I would also improve the presentation of the results, if possible.
Author Response
The study has improved a lot since its inception and now looks much more scientific and professional. For me, the underlying literature is still incomplete and I would also improve the presentation of the results, if possible.
The authors divide the above questions into the following 3 points:
Point 1: The study has improved a lot since its inception and now looks much more scientific and professional. For me, the underlying literature is still incomplete and I would also improve the presentation of the results, if possible.
Response 1: Thank you for your recommendation. PCT and nudge-related literature are the theoretical basis of this article, so the authors tries to increase nudge-related literature as much as possible. (Please see line 74-75, and line 99-102 red words)